# Application of 1-Hydroxy-4,5-Dimethyl-Imidazole 3-Oxide as Coformer in Formation of Pharmaceutical Cocrystals

**DOI:** 10.3390/pharmaceutics12040359

**Published:** 2020-04-15

**Authors:** Aneta Wróblewska, Justyna Śniechowska, Sławomir Kaźmierski, Ewelina Wielgus, Grzegorz D. Bujacz, Grzegorz Mlostoń, Arkadiusz Chworos, Justyna Suwara, Marek J. Potrzebowski

**Affiliations:** 1Centre of Molecular and Macromolecular Studies, Polish Academy of Sciences, Sienkiewicza 112, 90-363 Lodz, Poland; awroblew@cbmm.lodz.pl (A.W.); jsniech@cbmm.lodz.pl (J.Ś.); kaslawek@cbmm.lodz.pl (S.K.); ms@cbmm.lodz.pl (E.W.); achworos@cbmm.lodz.pl (A.C.); jmilczar@cbmm.lodz.pl (J.S.); 2Institute of Technical Biochemistry, Lodz University of Technology, Stefanowskiego 4/10, 90-924 Lodz, Poland; grzegorz.bujacz@p.lodz.pl; 3Department of Organic and Applied Chemistry, University of Lodz, Tamka 12, 91-403 Lodz, Poland; grzegorz.mloston@chemia.uni.lodz.pl

**Keywords:** imidazole *N*-oxides, barbituric acid, thiobarbituric acid, pharmaceutical cocrystals, mechanochemistry, solid state NMR, X-ray Diffraction

## Abstract

Two, well defined binary crystals with 1-Hydroxy-4,5-Dimethyl-Imidazole 3-Oxide (HIMO) as coformer and thiobarbituric acid (TBA) as well barbituric acid (BA) as Active Pharmaceutical Ingredients (APIs) were obtained by cocrystallization (from methanol) or mechanochemically by grinding. The progress of cocrystal formation in a ball mill was monitored by means of high-resolution, solid state NMR spectroscopy. The ^13^C CP/MAS, ^15^N CP/MAS and ^1^H Very Fast (VF) MAS NMR procedures were employed to inspect the tautomeric forms of the APIs, structure elucidation of the coformer and the obtained cocrystals. Single crystal X-ray studies allowed us to define the molecular structure and crystal packing for the coformer as well as the TBA/HIMO and BA/HIMO cocrystals. The intermolecular hydrogen bonding, π–π interactions and CH-π contacts responsible for higher order organization of supramolecular structures were determined. Biological studies of HIMO and the obtained cocrystals suggest that these complexes are not cytotoxic and can potentially be considered as therapeutic materials.

## 1. Introduction

Pharmaceutical cocrystals have recently received a great deal of attention because of their therapeutic importance and remaining challenges [1,2,3,4,5]. The commercial success of rationally designed drugs which have been introduced into the market in last decade, such as Steglatro^®^ (a molecular cocrystal of ertugliflozin and L-pyroglutamic acid), Odomzol^®^ (a cocrystal of sonidegib and phosphoric acid), Suglat^®^ (a cocrystal of ipragliflozin and L-proline), and Entresto1^®^ (a cocrystal of valsartan and sacubitril), has prompted “Big Pharma” companies to double their efforts regarding testing the compositions of new solid systems fulfilling the conditions whereby they may be classified as cocrystals.

According to the generally accepted definition, cocrystals are homogenous (single phase) crystalline structures which are made up of two or more components in a definite stoichiometric ratio where the arrangement in the crystal lattice is not based on ionic bonds (as with salts) [6].

The remarkable advantage of pharmaceutical cocrystals is their significant improvement in terms of physicochemical properties without compromising on therapeutic benefit [7]. It is well known that many Active Pharmaceutical Ingredients (APIs) suffer from low solubility and/or poor permeability [8,9]. Notably, up to 90% of new drugs receive a BCS II (Biopharmaceutics Classification System) category rating, meaning that they demonstrate low solubility and high permeability [10]. In the case of cocrystals, these parameters are usually significantly enhanced.

For scientists dealing with solid state matter and the formulation of new crystalline materials, the preparation of cocrystals with desired properties is usually a challenging task. Progress in this field has been made possible due to great achievements in crystal engineering which allow researchers to predict supramolecular interactions, resulting in new solid forms [11]. Generally, therapeutic cocrystals consist of two components: a biologically active organic compound and a complementary molecular coformer. In the pharmaceutical sciences, the correct choice of a coformer for a desired API is essential [12]. The library of available coformers which fulfill structural prerequisites is significant; however, only selected compounds can be considered to belong to the group of “pharmaceutically useful” ones [13,14,15]. The basic requirement for a suitable coformer is to be pharmaceutically acceptable, i.e., generally regarded as safe (GRAS) substances. Furthermore, coformers should be relatively cheap, with rather low molecular weight, and possess multiple API-binding sites which can be involved in the formation of strong intermolecular interactions [16].

The intention of our work was to introduce a new compound containing the imidazole core to the group of useful pharmaceutical coformers. In the current project, we selected 1-Hydroxy-4,5-Dimethyl-Imidazole 3-Oxide (**1**) 1-Hydroxy-4,5-Dimethyl-Imidazole 3-Oxide (HIMO) [17] (see Figure 1), which has never been tested before in therapeutic applications. This compound, bearing appropriate functional groups, is capable of forming ordered solid structures. Moreover, a strong motivation for its selection was the fact that numerous imidazole derivatives play an important role in organic chemistry, medicinal chemistry as well as the pharmaceutical sciences. Imidazole derivatives with histidine are common in nature, as basic amino acids. Being a part of the structure of protein, histidine is an essential amino acid which is crucial for the catalytic activity of many enzymes. Moreover, imidazole-based drugs exhibit antiviral [18,19,20], antitumor [21,22,23], bacteriostatic [24], and antiprotozoal [25,26] activities. They were reported to act as hypotensive agents [27] and selective inhibitors for some kinases [28,29]. Such unique properties prompted us to prepare new cocrystals, which can be also considered as unknown drug–drug binary systems.

The compound 1-hydroxy-4,5-dimethyl-imidazole 3-oxide (**1**) exists in the equilibrium of tautomeric forms [30]. In solution and in gaseous phase, it forms a mixture of the –OH (i.e., 1-hydroxyimidazole) and N→O (i.e., imidazole 3-oxide) tautomers existing in comparable amounts; this fact has been demonstrated by theoretical and experimental studies. The observed equilibria depend not only on the solvent polarity, but also on the composition of substituents attached to the imidazole ring, and their ability to form the inter- or intra- molecular bonds [31,32,33,34,35,36,37,38]. On the other hand, knowledge about the structure of **1** in the solid state is limited, and this makes this compound even more intriguing.

In the present study, thiobarbituric acid (TBA) (**2**) and barbituric acid (BA) (**3**) (Figure 1) were chosen as potential APIs. Both compounds are well known and have been described in numerous crystallographic and physico-chemical reports [38,39,40].

As highlighted above, the imidazole *N*-oxide **1** has never been tested as a pharmaceutical coformer. Thus, in view of its potential use, we present also biological studies performed for HIMO and its cocrystals with TBA and BA, respectively. Compounds **1**, **2**, **3** and new cocrystals were analyzed using solid-state NMR spectroscopy (SS NMR), single-crystal and powder X-ray Diffraction (XRD) techniques, as well as Differential Scanning Calorimetry (DSC).

## 2. Materials and Methods

The 1-hydroxy-4,5-dimethyl-imidazole 3-oxide (**1**) was obtained employing the procedure reported in the literature [17]. Compounds **2** and **3** were purchased from Sigma Aldrich (Poznan, Poland) and used without further purification. Reagent grade solvents were purchased from Polish Chemicals Reagents (POCH) company (Gliwice, Poland).

### 2.1. Preparation of Thiobarbituric Acid/1-Hydroxy-4,5-Dimethyl-Imidazole 3-Oxide (TBA/HIMO) Cocrystal

#### Method 1

Thiobarbituric acid (**2**) (14.5 mg, 0.10 mmol) and **1** (13.0 mg, 0.10 mmol) were dissolved in boiling MeOH and left for 8 days at room temperature for crystallization. After that, the precipitated TBA/HIMO cocrystals were filtered and used for further investigations.

#### Method 2

Thiobarbituric acid (**2**) (67.0 mg, 0.47 mmol) and **1** (60.0 mg, 0.47 mmol) were mixed with 0.15 mL of MeOH as a LAG and ground for 1 h in a ball-mill (agate jar—5 mL agate ball—5 mm; 25 Hz). After that, the desired cocrystals were removed from the jar and used for further investigation employing analytical techniques.

### 2.2. Preparation of Barbituric Acid/1-Hydroxy-4,5-Dimethyl-Imidazole 3-Oxide (BA/HIMO) Cocrystal

#### Method 1

The barbituric acid (**3**) (13.0 mg, 0.10 mmol) and **1** (13.0 mg, 0.10 mmol) were dissolved in boiling MeOH and left for 5 days at room temperature for crystallization. After that time, the precipitate of BA/HIMO cocrystals was filtered and used for further investigations.

#### Method 2

The barbituric acid (**3**) (60.0 mg, 0.47 mmol) and **1** (60.0 mg, 0.47 mmol) were mixed with 0.15 mL of MeOH as a LAG and ground for 2 h in a ball-mill (agate jar—5 mL agate ball—5 mm; 25 Hz). After that, the desired cocrystals were removed from jar and investigated employing analytical techniques.

### 2.3. NMR Spectroscopy

The NMR measurements were carried out using Bruker Avance III 600 and on a Bruker Avance III 400 spectrometers (Bruker GMbH, Karlsruhe, Germany). The Bruker Avance III 600 was equipped with 4 mm ^1^H/BB (^31^P–^15^N) CP/MAS probehead. The resonance frequencies for ^1^H, ^13^C, ^15^N were set at 600.13, 150.90, and 60.82 MHz, respectively. The ^13^C and ^15^N CP/MAS spectra were measured with a proton 90° pulse length of 4 μs, a 4–200 s repetition time and SPINAL64 decoupling (83 kHz amplitude). The ^1^H Very Fast MAS NMR spectra were recorded with a spinning speed of 60 kHz in a 1.3 mm zirconium rotor. Bruker Avance III 400 spectrometer experiments were performed for ^1^H, ^15^N and ^13^C at 400.15, 40.53 and 100.63 MHz resonance frequencies, respectively. The cross polarization magic angle spinning (^13^C CP MAS) spectra were recorded with a proton 90° pulse length of 4 μs.

In typical CP/MAS experiments, samples were packed into a 4 mm ZrO_2_ rotor and spun at a rate of 8–12 kHz. The 1024 transients were acquired. FIDs were accumulated using time domain size of 3.6 K data points. Adamantane (resonances at 38.48 and 29.46 ppm) was used as a secondary ^13^C chemical-shift reference from external tetramethylsilane (TMS). For ^15^N, a chemical shift calibration glycine signal of δ = 36.6 ppm was used as a secondary reference.

### 2.4. Single Crystals X-ray Diffraction Measurements

The HIMO, TBA/HIMO and BA/HIMO crystals were obtained by crystallization from methanol. Single crystal diffraction experiments were carried out using Oxford SuperNova single-crystal diffractometer (Agilent Technologies, Yarnton, UK) with micro-source CuKα radiation (λ + 1.5418Å) and a Titan detector at room temperature. These conditions were chosen deliberately to show molecular dynamics similar to those present in NMR experiment. The absorption correction was performed based on the crystal shape, orientation and absorption coefficient. Diffraction data collection, cell refinement, data reduction and absorption correction were performed using the CrysAlis PRO software (Oxford Diffraction). Structures were solved by direct method SHELXS and then refined using the full-matrix least-squares method, i.e., SHELX 2015, implemented in the OLEX2 package. The crystals of TBA/HIMO showed pseudo merohedral twinning, which is rare for triclinic systems, possible due to similar *a* and *b* unit cell parameters and α and β cell angles. The refinement was initially performed using one lattice component from diffraction data, and later, the second lattice component was introduced (HKLF 5), which significantly improved the statistical parameters. The hydrogen atoms were set geometrically and refined as riding with the thermal parameter set to 1.2 of the thermal vibration of the parental atom. The dislocated hydrogen atoms were found on the difference Fourier map and refined with 0.5 occupancy for both alternative positions with geometrical restraints and thermal parameters equal to 1.5 of the thermal vibration of the parental atom. These structures were validated by CheckCif (http://checkcif.iucr.org) and deposited in the Cambridge Crystallographic Data Centre (CCDC) under accession numbers 1967291, 19672932 and 1967293 for, TBA/HIMO, BA/HIMO and HIMO and crystals, respectively.

### 2.5. X-ray Diffraction of Powders

Powder X-ray diffraction experiments were carried out using a Bruker D8 Advance diffractometer (BRUKER AXS, Inc., Madison, WI, USA) equipped with a LYNXEYE position sensitive detector using CuKα radiation (λ = 0.15418 nm). The data were collected at room temperature in the Bragg–Brentano (θ/θ) geometry, between 3° and 60° (2θ) in a continuous scan using 0.03° steps, under standard laboratory conditions without prior preparation of the measured samples.

### 2.6. Differential Scanning Calorimetry (DSC) Measurements

DSC measurements were recorded using a DSC 2920, (TA Instruments, New Castle, DE, USA). The heating processes investigated for the TBA/HIMO and BA/HIMO cocrystals were performed in a muffle furnace (SM-2002) manufactured by Czylok (Jastrzebie Zdroj, Poland). The measurements were carried out in temperature range of 0 °C–300 °C. The heating rate was 10 K/min. The sample mass was 1.7040 mg for TBA/HIMO and 1.8200 mg for BA/HIMO.

### 2.7. Biological Studies of HIMO, TBA/HIMO and BA/HIMO

The cytotoxic impact of the tested compounds on the viability of cells after concentration-dependent treatment was determined by standard MTT [3-(4,5-dimethylthiazol-2-yl)-2,5-diphenyltetrazolium bromide] assay. Experiments were performed with cancer (HeLa–cervical cancer, K562–chronic myelogenous leukemia) and noncancerous (fibroblasts, 293T–derived from human embryonic kidney) human cell lines. A colorimetric assay was used to measure the cell viability through the metabolism of tetrazolium salt into formazan by mitochondrial dehydrogenase in living cells. In brief, the cells were plated into 96-well transparent plates (Nunc) at a density of 7 × 10^3^ cells per well in 200 μL of fresh RPMI or DMEM medium (supplemented with 10% fetal bovine serum and 1% penicillin and streptomycin). After overnight incubation (37 °C, 5% CO_2_), the medium was replaced for the control and the wells containing various concentrations (1 μM, 10 μM, 50 µM and 100 µM) of the tested compounds. After a further 48 h of incubation under the same conditions, 25 µL of MTT solution (5 mg/mL) was added to each well. After a subsequent 2 h of incubation, 95 μL of lysis buffer (20% SDS, 50% aqueous DMF, pH 4.5) was added. Afterward, the plates were mixed on a microplate shaker (2 h at room temperature) to dissolve the formazan. Then, the optical density (OD) was measured by a microplate reader at 570 nm with a reference wavelength of 630 nm. Percent cell viability relative to the control was calculated as cell viability (%) = (OD treatment/OD untreated control cells) × 100%. The data represent the mean values from five repeats from three independent experiments.

### 2.8. Solubility Measurements of BA, TBA, TBA/HIMO and BA/HIMO

Solubility measurements were performed by preparing supersaturated solutions of the tested cocrystals (BA/HIMO, TBA/HIMO) and pure BA and TBA in Milli-Q water (pH 5.7), simulated gastric fluid without pepsin (SGFsp, pH 1.2) and ethanol. SGFsp was prepared by adding 2 g NaCl and 7 mL concentrated HCl to 1000 mL distilled water. The obtained suspensions were stirred at room temperature for 24 h. After filtration through a 0.45 μm PTFE syringe, the filtrates were diluted with water to obtain an appropriate concentration (in the range of UPLC-MS calibration).

The concentration of BA and TBA was determined by an ACQUITY UPLC I-Class chromatography system coupled with SYNAPT G2-Si mass spectrometer equipped with an electrospray source and quadrupole-Time-of-Flight mass analyzer (Waters Corp., Milford, MA, USA). Acquity BEH_TM_ C18 column (100 × 2.1 mm, 1.7 μm), maintained at 45 °C for the chromatographic separation of analyte. A gradient was employed with the mobile phase combining solvent A (1% formic acid in water) and solvent B (1% formic acid in acetonitrile) as follows: 3% B (0–0.5 min), 3–97% B (0.5–1.5 min), 97–97% B (1.5–1.8 min), 97–3% B (1.8–1.9 min) and 3–3% B (1.9–3.5 min). The flow rate was 0.40 mL/min, and the injection volume was 2 μL.

For mass spectrometric detection, the electrospray source operated in negative resolution mode. The optimized source parameters were as follows: capillary voltage 2.8 kV; cone voltage 25 V; desolvation gas flow 900 L/h with a temperature of 450 °C; nebulizer gas pressure 6.5 bar; and source temperature 110 °C. Mass spectra were recorded over an *m*/*z* range of 100 to 1200. Mass spectrometer conditions were optimized by direct infusion of the standard solution. The system was controlled using the MassLynx software (Version 4.1), and data processing (peak area integration, construction of the calibration curve) was performed using TargetLynxTM (Waters Corp., Milford, MA, USA).

The initial stock calibration solutions of BA and TBA were prepared at a concentration of approximately 10 mg/mL in water and stored at 4 °C. The stock solutions were diluted (with water) to obtain the target concentrations. The calibration curves were prepared at seven different concentrations of BA and TBA. The calibration curves were linear over a concentration range from 10 ng/mL to 1500 ng/mL with the correlation coefficients of >0.994 and 0.992, respectively, for BA and TBA.

The concentrations determined for BA and TBA (in pure form and in the tested cocrystals) were reported as the average of two replicated experiments for each sample. The injections for each sample were repeated four times.

## 3. Results

### 3.1. Solid State NMR Studies of HIMO

One of the aims of the present project was the comparison of two methods of preparing cocrystals, i.e., classic cocrystallization of components with solvent (solvents) use and solvent free ball milling. For the monitoring of the processes employing the ball mill method, knowledge about the structure and properties of HIMO (1) in the solid state was crucial. In our structural studies of 1 in condensed matter, we started with a ^13^C CP/MAS NMR experiment performed with a spinning rate of 8 kHz at ambient temperature in a 4 mm zirconium rotor. Figure 2a shows spectrum of the sample purified by precipitation. The unexpected feature of the spectrum was its very high resolution of resonances, which is typical for highly crystalline samples with very well organized crystal lattice. Similarly, high quality spectra are usually recorded for molecules which have crystallized in multiple steps. Going further and analyzing the number of resonances both in aromatic (imidazole) and aliphatic (methyl) regions, we concluded that 1 crystallizes during precipitation in space group with Z’ equal 2. The four, clearly cut methyl signals provide unambiguous proof to confirm this hypothesis.

The ^15^N CP/MAS spectrum (Figure 2b) confirms the presence of more than one molecule in the asymmetric part of the unit cell. Three clearly cut signals found at δ = 208.0, δ = 214.7 and δ = 216.0 ppm differed in intensity. We assumed that the larger intensity of the middle peak (ca. 215 ppm) resulted from the superposition of two resonance signals representing the A and B molecules.

The ^1^H NMR spectroscopy is an interesting source of information. For this measurement, we employed a unique technique called Very Fast Magic Angle Spinning (VF MAS). The “very fast” (VF) regime, i.e., at more than 50 kHz, is obtained using commercially available 1.3 mm rotors. This frequency exceeds the strength of the homonuclear ^1^H dipolar coupling, and is therefore expected to show a new regime for spin dynamics. In many cases, the ^1^H NMR spectra recorded under VF MAS conditions showed acceptable resolution and the protons positions (chemical shifts) could be correctly assigned.

The ^1^H VF MAS spectrum (Figure 2c) recorded with spinning rate 60 kHz displays three groups of signals representing protons of methyl residues (δ_1H_ = 1.36 ppm), protons bonded to methine carbon C2 (δ_1H_ = 9.17 ppm) and protons of hydroxyl residue (δ_1H_ = 17.71 ppm). The chemical shifts of *H*C(2) and O*H* are not straightforward. In the first case, the strong deshielding effect was due to the aromatic character of imidazole strengthened by the “zwitterionic” nature of the adjacent N→O unit. In general, the chemical shift at around 9 ppm is characteristic for positions called “acidic”. The value of δ_1H_ for O*H* suggests that this residue is involved in a strong hydrogen bonding interaction.

### 3.2. Formation and Structural Studies of TBA/HIMO and BA/HIMO Cocrystals

#### 3.1.1. SS NMR Studies of TBA/HIMO Cocrystal

The TBA **2** and BA **3**, as well as their derivatives, commonly known as barbiturates, play an important role in pharmaceutical industry because of their hypnotic, sedative, anticonvulsant, antimicrobial, anesthetic, anticancer and antitumor properties [41]. Due to the unique structure of **3** and their ability to form keto–enol tautomers, it is widely used as a valuable building block in organic synthesis and advanced materials including cocrystals [42,43].

In our project, the TBA/HIMO cocrystals were obtained by the cocrystallization of equimolar amounts of TBA and HIMO dissolved in boiling methanol. When the solution was cooled down and kept at ambient temperature for 8 d in a closed vessel, the expected crystals were efficiently formed. After separation of the TBA/HIMO cocrystals, they were dried in air and measured employing ^13^C CP/MAS, PXRD and DSC techniques (Figure 3). All experiments confirmed the formation of new crystallographic structures. It is worth noting that in the DSC profile (Figure 3c), a strong exothermic peak corresponding to thermal decomposition, typical for high-energetic cocrystals, was observed.

It is apparent that in the case of wet methods of cocrystal formation, which are based on the dissolving of starting components, the crystallographic form of applied ingredients is irrelevant. In the case of mechanochemical methods (e.g., ball milling), the structural effects (crystal structure, polymorphism, tautomerism, intermolecular contacts) can affect the process of cocrystal formation. It seems that TBA, for which a rich collection of polymorphs and one hydrated form have been isolated, is an appropriate candidate with which to test the correlation between crystallographic form and the ball milling process. In the crystal lattice, TBA exists in enol form, keto isomer, as well as a mixture of keto/enol forms, each of which can be easily recognized by ^13^C CP/MAS experiments (see [43]). Depending on the solvent used for the crystallization, it is possible to obtain a suitable form of TBA.

The structure, tautomerism and crystal forms of TBA were investigated by Chierotti et al. [44]. Three different forms crystallized from different solvents were also used as starting materials in the present study. TBA(I) was obtained using MeCN as a solvent for crystallization. This sample exists as tautomeric form A (Scheme 1).

In this form, the C(O)NH moieties on all molecules are either involved in the typical hydrogen-bonding ring motif, with formation of a dimer, or in the formation of hydrogen-bonded chains. TBA (II) was obtained by slow evaporation of a hot ethanolic solution. TBA (II) crystallizes as a tautomer B and forms a type of hydrogen-bonded zigzag chain. Recrystallization of TBA from water leads to the formation of hydrated form III, presented as B, with 1.5 water molecules per formula unit. The acid molecules form a chain through two types of hydrogen-bonding rings involving C(O)NH and C(S)NH moieties.

Figure 4 (left column, a–c) shows the ^13^C CP/MAS spectra for the physical mixture of two components, i.e., TBA in different forms (I-III) and HIMO, in a 1:1 ratio. It is clear that the intensity of the signals does not reflect the content of the mixture. This is because the relaxation times and efficiency of the cross-polarization for both compounds were significantly different. The TBA/HIMO cocrystals were obtained mechanochemically in the presence of MeOH added as a LAG. Figure 4 (right column, d–f) shows the ^13^C CP/MAS spectra after one hour of grinding in a Mixer Mill MM 200 equipped with a 5 mL agate jar and 5 mm diameter balls at an oscillation rate of 25 Hz. A comparative analysis of the spectra proved that during the grinding process, in each case, the formation of cocrystals was observed, and the final product was always very similar. The only difference was observed for TBA (III); in that case, a small amount of unreacted HIMO was detected in the crude mixture. Apparently, the 1 h grinding was not sufficient to complete the expected transformation.

The ^13^C CP/MAS results suggest that during the mechanosynthesis, each TBA form yields the same product in the presence of HIMO. This hypothesis was further confirmed by ^15^N CP/MAS experiments. As in the case the of ^13^C NMR, the left column depicts the spectra of physical mixtures of TBA I-III/HIMO (Figure 5a–c). The TBA I crystallized from MeCN (Figure 5a) is characterized by two sharp peaks at δ = 171.7 and 167.2 ppm, respectively. For TBA II, the chemical shifts were found to be at δ = 169.8 ppm and 160.0 ppm, while for TBA III, the chemical shits were at δ = 174.1 and 157.2 ppm. The distinction in relative intensities of the HIMO versus the TBA ^15^N signals depended on differences in relaxation times of TBA. The right column (Figure 5d–f) displays the spectra for the obtained cocrystals. An inspection of the ^15^N CP/MAS data shows apparent resemblances. The number of resonances in the isotropic part suggests that in the asymmetric unit, at least two pairs of TBA/HIMO molecules exist (Z’ = 2).

The ^1^H VF/MAS measurements offer an additional source of information for the progress of cocrystal formation. Figure 6 shows the spectra for the mixture of both components (Figure 6a,c,e) and samples after 1 h grinding (Figure 6b,d,f). The ^1^H chemical shifts for OH and NH protons located in the region 18–10 ppm provide evidence for the very complex hydrogen bonding network in the crystal lattice. The blue vertical line represents the position of OH proton for pure, unreacted HIMO. The small shift of this signal is a probe sensing the course of grinding process. Based on the analysis of spectra for ball milled samples, one can conclude that in experiments with TBA I and TBA II, the transformation was quantitative, while in case of TBA III (crystallized from water), the part of HIMO was not involved in the formation of cocrystals. This observation is consistent with both ^13^C and ^15^N analysis.

#### 3.1.2. SS NMR Structural Studies of BA/HIMO Cocrystal

In the next step, employing wet and solid state methods, we tested the ability of HIMO to form cocrystals with barbituric acid (BA), an analog of TBA. As before, the cocrystallization of BA with HIMO in MeOH led to the formation of the desired BA/HIMO cocrystals. Its structure was confirmed by ^13^C CP/MAS, PXRD and DSC measurements (Figure 7).

Analyzing the reactivity of BA in mechanochemical procedures, it has to be noted that this compound forms different polymorphs, two anhydrous forms (form I and II, see Scheme 2) and a dihydrate phase. They present a trioxo structure, as confirmed by X-ray diffraction analysis. Recently, Chieriotti and coworkers proved the existence of unstable trihydroxyl and keto-enol tautomers of BA in the crystal lattice [45].

Figure 8a shows a mixture of BA and HIMO in a molar ratio of 1:1, crystallized from MeOH (labeled in the literature as Form II). The ^13^C CP/MAS spectra for BA form II have already been reported [44,45]. Two CH_2_ peaks located at 39.1 and at 41.1 ppm represent two independent molecules in the asymmetric unit cell where the CH_2_ moiety can be in or out of the plane of the ring in a half chair conformation. The resonances at 151.9, 170.4 and 171.6 (shoulder) ppm were associated with the C2, C6, and C4 carbon atoms of the carbonyl groups, respectively. The assignment of ^13^C NMR resonances for 1 is presented in Section 3.1. Figure 8c displays the sample prepared mechanochemically (Mixer Mill MM 200 equipped with a 5 mL agate jar and 5 mm diameter ball at an oscillation rate of 25 Hz). The ^13^C CP/MAS pattern shows changes and provides proof for the formation of a new species. The new signal at 79.5 ppm, which is characteristic for keto/enol-form, is apparent.

In extension of this study, we investigated crystals of BA containing two water molecules in the lattice, and the susceptibility of this form to yield cocrystals. Recently, King and coworkers reported on detailed structural studies on, and the unusual thermal phase behavior of, barbituric acid (BA) dihydrate [46]. Our results prove that during grinding at ambient temperature, this form is stable and does not undergo further transformation.

Figure 8b shows the ^13^C CP/MAS spectrum of the mixture BA·2H_2_O with HIMO in 1:1 molecular ratio. The peak at δ = 40 ppm is characteristic for the keto form of BA. The cocrystal was prepared by grinding in the Mixer Mill MM 200. In this case, we did not use LAG, assuming that crystalline water could be a substitute. After 1 h ball-milling, the process of BA/HIMO cocrystal formation was complete. The registered ^13^C NMR spectrum, like those shown in Figure 8c, confirms the structure of product.

The presence of the keto-form of BA further confirmed the validity of the ^15^N CP/MAS experiment. The spectrum of BA (Figure 9a) with one NMR signal located at δ = 150.5 ppm is consistent with data published by Chierotti et al. [45]. In the case of BA dihydrates, signals at δ = 150.5 and 153.6 ppm were observed (Figure 9b); Figure 9c shows the ^15^N CP/MAS spectrum of BA:HIMO cocrystal. Analysis of this spectrum suggests that ^15^N resonance representing the BA component was upfield shifted by ca. 10 ppm (δ = 140.2 ppm) compared to pure BA, while ^15^N representing HIMO showed a split signal at δ = 208.0 ppm. Figure 9d displays the ^1^H NMR VF/MAS spectrum of HIMO and BA (ratio 1:1) after 2 h of grinding recorded with a spinning rate 60 kHz. The resolution of this spectrum makes the assignment of proton signals unambiguous. The first striking difference is the lack of the OH signal at δ_1H_ = 17.7 ppm, which, in pure sample 1, represents strong O-H···O hydrogen bonding and the appearance of new signals at δ = 15.7, 14.8 and δ = 9.6 ppm. The collected NMR data are a valuable source of information about the organization of new materials (HIMO and BA in a ratio of 1:1 after 2 h of grinding) and about the tautomeric form of BA.

### 3.3. X-ray Structure of HIMO and TBA/HIMO, BA/HIMO Cocrystals

Pure HIMO crystallizes in the triclinic system in the centrosymmetric P-1 space group. The asymmetric unit contains two molecules of HIMO (Z’ = 2). The crystal structure is displayed in Figure 10. Table 1 shows the experimental details and structural information.

The supramolecular array depicted in Figure 11 shows two structural features which are worthy of mention. Firstly, there are two strong hydrogen bonds with O···O distances equal 2.437 Å and 2.446 Å between molecules, creating asymmetric units and molecules related by a symmetry operation. These contacts are responsible for the formation of the chain structure. In both cases, the position of the hydrogen atom is averaged between both oxygen acceptors at two locations with half occupancy. The second feature seen in Figure 11 is a π–π interaction which is an additional factor stabilizing the crystal structure.

The crystal structure and molecular packing of TBA/HIMO is shown in Figure 12. The experimental details and structural information are collected in Table 1. The TBA/HIMO crystallizes in the triclinic system in the centrosymmetric P-1 space group. The independent unit contains two molecules of TBA and two molecules of HIMO. Both components in the crystal lattice are connected by hydrogen bonds forming a unique supramolecular structure. The leading motif is created by the planary located TBA molecules forming ribbons (flat chains), rotated such that the thiocarbonyl groups point into the opposite directions. These chains are bonded by strong hydrogen bridges between C_C_(2)=O_C_(1)···H_D_(2)–N_D_(2) and C_C_(4)=O_C_(2)···H_D_(1)–N_D_(1), and further by C_D_(4)=O_D_(2)···H_C_(1)–N_C_(1) and C_D_(2)=O_D_(1)···H_C_(2)–N_C_(2). TBA chains are connected by the HIMO molecules to create a plane structure. The methyl groups of HIMO from two layers are pointing into the plane interface. The interaction in other interfaces between planes is created by π stacking interactions.

Figure 13 shows the supramolecular pattern for TBA/HIMO in the crystal lattice. Clearly visible characteristic motifs are sheets formed by TBA molecules. These planes are created from the ribbons described above. The distances between O_C_(1)···N_D_(2) and O_C_(2)···N_D_(1) were found to be 2.854 Å and 2.839 Å, respectively. The analogous O_D_(2)···N_C_(1) and O_D_(1)···N_C_(2) lengths are equal to 2.850 Å and 2.886 Å. The oxygen atoms O1 and O2 of molecules A and B are involved in bifurcated hydrogen bonding with neighboring HIMO molecules. The strength of these bonds is defined by short O···O contacts equal 2.432 Å, 2.598 Å and 2.503 Å, 2.595 Å, respectively. The imidazole *N*-oxide molecules act as a link between adjoining TBA sheets during the formation of hydrogen bridges. As we postulated, at this stage, one of the hydrogens of the methylene group is transferred to the oxygen, then to HIMO oxygen forming intermolecular bonding. The second oxygen acts as an acceptor by interacting with OH residue in HIMO. In such an arrangement, carbonyl C=O groups, which are involved in strong hydrogen bonding, disturb the tautomeric nature of TBA. The TBA loses its pure keto tautomeric form and exists in keto-enol C–O···H–O–N form. Such a complex transfer process makes it difficult to refine the position of the hydrogen in the bridges but position of peak on electron density map shows that is predominantly attached to oxygen in HIMO.

Finally, the supramolecular structure of TBA/HIMO is supplemented by CH···π interactions between the methyl groups of HIMO and the ring of TBA. The average distance between the planes of the HIMO and TBA sheets, measured as the distance between the CH_3_ atom and the center of the TBA ring, is ca. 4 Å. The imidazole rings are in a stacked arrangement, and are twisted with respect to the TBA plane. It is worth noting that the C=S moiety, in this crystal form, interacts with the π-electrons of imidazole. The distance between the sulfur and the center of the imidazole ring is 3.5 Å.

Figure 14 shows the crystal structure and molecular packing of the BA/HIMO cocrystals. The experimental details and structural information are collected in Table 1. The BA/HIMO sample crystallizes in the triclinic system in the centrosymmetric P-1 space group. The asymmetric unit contains one molecule of BA and one of HIMO (Z = 2). The supramolecular structure is depicted in Figure 15. The motifs seen in the TBA/HIMO structure are also observed for BA/HIMO. The first similar pattern is the ribbon structure (flat chains) formed by barbituric acid. The hydrogen bonding network is similar to the one observed before. The layers are formed by BA ribbons located perpendicularly to the plane and stabilized by π stacking interactions. The HIMO molecules bridge the layers of BA such that protonated oxygens create hydrogen bonds with keto groups of BA. The lengths of N-H···O=C bonds were found to be 2.869 Å and 2.912 Å, respectively. The C(1)=O(1) and C(4)=O(3) groups are involved in the formation of bifurcated hydrogen bonds. The counter partner in these bridges is HIMO, which acts as a staple connecting the BA sheets located in parallel planes. The O···O contacts are equal to 2.474 Å and 2.593 Å. The keto-enol form is created according to the mechanism described above for TBA, and is also effective for BA/HIMO cocrystals. The evidence confirming the hypothesis that the tautomeric form is forced by hydrogen bonds is found within the analysis of the C-O bond lengths for BA. For C(2)–O(2), this length is 1.222 Å, while for C(1)=O(1) and C(4)=O(3), it is 1.283 Å and 1.276 Å, respectively. It is interesting to note that for pure anhydrous BA, existing in keto form, these bond lengths are 1.229 Å, 1.222 Å, 1.189 Å for C(2)=O(2), C(1)=O(1) and C(4)=O(3), respectively.

The sheets formed by BA are located in parallel planes with a distance of ca. 4 Å. The HIMO clips next to the C(1)=O(1) and C(4)=O(3) BA positions, responsible for creating of structural network, are oriented in opposite directions. Such an orientation allows the BA sheets to join in a “zipper” type mechanism.

Concluding this part, the presented X-ray results are consistent with a publication by Braga and coworkers, who, while studying the gas–solid reactions between the different polymorphic modifications of barbituric acid and amines, reported similar structural motifs with complex hydrogen bonding networks [47].

### 3.4. Cell Cytotoxicity and Solubility of Tested Compound (HIMO and Its Cocrystals)

As highlighted in the Introduction, 1-hydroxy-4,5-dimethyl-imidazole 3-oxide (**1**) has never been considered as a coformer in the formation of pharmaceutical cocrystals. Among the main problems were a lack of biological tests and limited knowledge about its cytotoxicity.

In view of its potential medical application, cell viability should to be tested to exclude potential toxicity. The cytotoxicity of the studied compounds was measured by the MTT assay method against HeLa (cervical cancer), K562 (chronic myelogenous leukemia) and noncancerous, 293T (derived from human embryonic kidney). Figure 16 shows the cellular viability after concentration-dependent treatment; the results indicate no significant toxicity in the case of incubation with the tested compounds in the 1–100 μM concentration range. The viability of cells was at a level of 80–100 percent, and only in the case of the highest concentration of the first cocrystal (BA/HIMO) did the survival rate decrease to 70% with HeLa cells. The results for noncancerous 293T were similar to those for HeLa cells. The obtained data indicate that all the tested compounds are suitable candidates for use in live cancer and noncancerous cells at concentrations of 1 μM to 100 µM.

Figure 17 shows the solubility of pure TBA and BA samples versus the solubility of the studied cocrystals. The measurements were carried out in three media: water, SGFsp and EtOH. It can be concluded that in water (pH 5.7), the solubility of BA/HIMO cocrystal is slightly better compared to that of BA. In the case of TBA and TBA/HIMO, the relationship is reversed. The solubility of the cocrystals is reduced by over 30%. A cocrystal analysis in simulated gastric fluid without pepsin (SGFsp, pH 1.2) revealed a decrease in solubility for both binary systems compared to TBA and BA. For TBA/HIMO, solubility is reduced by ca. 17%, while for BA/HIMO, it was reduced by about 33% compared to the same cocrystals in water. In EtOH, the solubility of BA is higher compared to that of the cocrystal, while that of TBA and TBA/HIMO is comparable. Comparing the APIs and cocrystals, an increase in solubility was not observed. Usually, increased solubility is the first indication to begin the preparation of cocrystals, because this parameter has an influence on drug pharmacodynamics and the efficiency of treatment. In the case of TBA/HIMO and BA/HIMO, only a minute increase of solubility was noticed for BA/HIMO cocrystals in water. However, while solubility is an important aspect, this is not an arbitrary factor justifying the use of cocrystals in drug delivery. Other functions, for instance, the protection of sensitive APIs from the environmental effects or controlling the delivery of drugs to specific points in the body are important as well. Also relevant from a drug delivery point of view is providing more than one active pharmaceutical ingredient. This seems to be an attractive feature in the case of imidazole *N*-oxide 1 (HIMO).

## 4. Conclusions

This study demonstrates the first application of HIMO as a useful coformer for the preparation of pharmaceutical cocrystals. As API references, thiobarbituric (TBA) and barbituric (BA) acids were employed. Two different methods for the cocrystals preparation were applied: 1) cocrystalization from MeOH used as a solvent, and 2) the grinding of solid components in a ball mill. In both cases, the desired cocrystals (TBA/HIMO and BA/HIMO) were obtained. An advantage of the ball-mill method is that in a classic, “wet” method, a substantial volume of organic solvents has to be used, and the procedure is relatively long (i.e., 8 days). Another important advantage of the ball milling method is its remarkable reduction of the reaction time required to complete the preparation of the cocrystals. Moreover, the ball milling procedure does not require toxic solvents, which is an important feature in view of economic and ecological concerns. The application of diverse analytical methods such as solid-state NMR spectroscopy, X-ray crystallography of single crystals and powders, and IR spectroscopy allowed us to perform an extended structural analysis of the obtained cocrystals. Solid state NMR techniques were used to monitor the progress of cocrystal formation. In addition, it allowed us to evaluate whether the polymorphic and tautomeric forms of the used components influenced the kinetics of the cocrystal formation and the structures of the final products. It was demonstrated that in each case, identical cocrystals formed, irrespective of the applied method (solvent versus ball milling). However, in the latter procedure, TBA hydrate requires a longer processing time.

The biocompatibility for all investigated compounds was also investigated. Experiments performed with cancer (HeLa, K562) and noncancerous (293T) human cell lines suggest that none of tested compounds is toxic. The obtained results strongly suggest that both TBA/HIMO and BA/HIMO cocrystals can be considered potential candidates for pharmaceutical application at a wide range of concentrations, i.e., varying from 1 μM to at least 50 µM. The solubility of the obtained cocrystals was tested in three media. Although only a small increase of solubility was observed for BA/HIMO in water, it seems obvious that the other characteristics of the studied cocrystals (protection of APIs and providing more than one API) may be useful for the development of new drug delivery systems based on 1-hydroxy imidazole 3-oxide derivatives.

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
