# Peer review of "Application of 1-Hydroxy-4,5-Dimethyl-Imidazole 3-Oxide as Coformer in Formation of Pharmaceutical Cocrystals"

_pharmaceutics, 2020, doi:10.3390/pharmaceutics12040359_

Round 1
Reviewer 1 Report
Review from Manuscript ID: pharmaceutics-761230
Summary: The paper describes 2 cocrystals form TBA and BA with a coformer (HIMO). Several solid-state techniques are employed to study the cocrystals with emphasis on ss NMR.
Although the authors state that the objective of the paper is to test the HIMO as a potential coformer for pharmaceutical applications, the objective seems to be to use NMR and XRD to characterize solids. The objective of the paper is not clear. Is it to form pharmaceutical relevant cocrystals? It seems that in the end, the cocrystals obtained cannot be used to improve the solubility of this two APIs. If the objective was to test HIMO as a potential coformer, more APIs should be testes.
Detailed comments:
Line 39: What to the authors mean by mainly in stoichiometric ratio? The definition of cocrystals by the European Medicine Agency is: “cocrystals are homogenous (single phase) crystalline structures made up of two or more components in a definite stoichiometric ratio where the arrangement in the crystal lattice is not based on ionic bonds (as with salts).”
Line 165: The DSC description lacks information regarding temperature range, temperature rate and sample mass.
Figure 3: In this figure the comparison between the physical mixture and obtained cocrystals needs to be present. Otherwise is not clear that a new phase was obtained.
Figure 3. I do not understand the DSC thermogram. The thermogram presents just a single exothermic peak that is referred a thermal decomposition. Where is the melting? It decomposes in the solid state. Please clarify this and add the DSC thermograms of the parent compounds.
Figure 3: The XRPD diffractogram appear to have a broad baseline characteristic of an amorphous compound. Can you comment on this?
Figure 3. It would also be beneficial to the work to show the comparison between the cocrystals obtained by solvent evaporation and by grinding. Was the same form obtained by the different methods?
How can the authors be sure that cocrystals were formed and not salts, for example?
Line 334: I believe that is Figure 5 and not 4.
Figure 7: The same question that I made for Figure 3 can also be made here.
Line 553: Why did the authors determined the solubility in ethanol and Ethyl Acetate? Those two solvents are not relevant for pharmaceutical application. Moreover, the solubility should be made in a biorelevant medium at a pH close to the one present in the site of absorption of the APIs. None of that is accounted for in this work.
Line 554: The authors state that solubility is important, however cocrystals can be use for other purposes. Do the APIs used in this work need protection? Also, if the objective was to improve solubility, the solubility values of the APIs should eb stated in the introduction.
Author Response
Reviewer 1
Summary: The paper describes 2 cocrystals form TBA and BA with a coformer (HIMO). Several solid-state techniques are employed to study the cocrystals with emphasis on ss NMR. Although the authors state that the objective of the paper is to test the HIMO as a potential coformer for pharmaceutical applications, the objective seems to be to use NMR and XRD to characterize solids. The objective of the paper is not clear. Is it to form pharmaceutical relevant cocrystals? It seems that in the end, the cocrystals obtained cannot be used to improve the solubility of this two APIs. If the objective was to test HIMO as a potential coformer, more APIs should be testes.
Answer: We fully agree with comment of Referee. Moreover, it is consistent with our current projects. Actually different API’s are tested (including linezolid and apremilast) as systems for formation of new cocrystals with HIMO. Hopefully we will finish these studies soon.
Detailed comments:
Line 39: What to the authors mean by mainly in stoichiometric ratio? The definition of cocrystals by the European Medicine Agency is: “cocrystals are homogenous (single phase) crystalline structures made up of two or more components in a definite stoichiometric ratio where the arrangement in the crystal lattice is not based on ionic bonds (as with salts).”
Answer: It is changed in the revised version
Line 165: The DSC description lacks information regarding temperature range, temperature rate and sample mass.
Answer: The requested information is added in revised version of manuscript.
Figure 3: In this figure the comparison between the physical mixture and obtained cocrystals needs to be present. Otherwise is not clear that a new phase was obtained.
Answer. Thank you for this comment. For better clarity we decided to add the Supporting Information. In SI we show the calculated diffractograms for three forms of TBA used in our project employing crystallographic data published by Chierotti et al. Diffractograms for TBA sample crystallized in acetonitrile, ethanol, water as well HIMO and TBA/HIMO are depicted in Figure S1.
Figure 3. I do not understand the DSC thermogram. The thermogram presents just a single exothermic peak that is referred a thermal decomposition. Where is the melting? It decomposes in the solid state. Please clarify this and add the DSC thermograms of the parent compounds.
Answer. We understand the reviewer's embarrassment. The DSC profile shown in Figure 3 is not typical for simple organic compounds but is typical for highly energetic materials (EM). Matzger and coworkers have reported number of cocrystals fulfilling conditions for EM. See for instance papers “Achieving Balanced Energetics through Cocrystallization” Michael K. Bellas , Adam J. Matzger Angew. Chem. Int. Ed. 2019, 58, 17185 –17188, and “A melt castable energetic cocrystal” Jonathan C. Bennion,a Zohaib R. Siddiqia , Adam J. Matzger Chem. Commun., 2017, 53, 6065—6068. In this approach compounds possessing oxidizing organic functional groups (NO2, ONO2, etc.) are used as coformers. DSC profiles for cocrystals with 4-amino-3,5-dinitropyrazole (ADNP) used as coformer (Matzger works) show very similar thermograms as those reported in our work..
Figure 3: The XRPD diffractogram appear to have a broad baseline characteristic of an amorphous compound. Can you comment on this?
Answer. The solid state NMR studies did not confirm the presence of amorphous phase. The broad baseline in diffractogram is result of technical problem related with XRDP experiment . For this measurement we have had not enough material for covering whole surface of plate. For this reason the baseline is not perfect.
Figure 3. It would also be beneficial to the work to show the comparison between the cocrystals obtained by solvent evaporation and by grinding. Was the same form obtained by the different methods?
Answer. We agree with this comments. The appropriate 13C CP/MAS spectra for samples obtained by solvent evaporation and by grinding are attached in SI (Figure S2)
How can the authors be sure that cocrystals were formed and not salts, for example?
Answer. It is good point. Very recently this problem was discussed in paper “Selective Synthesis of a Salt and a Cocrystal of the Ethionamide−Salicylic Acid System” by Davide Bernasconi, Simone Bordignon, Federica Rossi, Emanuele Priola, Carlo Nervi,Roberto Gobetto, Dario Voinovich, Dritan Hasa, Nghia Tuan Duong, Yusuke Nishiyama, and Michele R. Chierotti.
In many cases, the main distinction between salts and cocrystals depends on whether a complete proton transfer has occurred or not along the axis of a hydrogen bond (HB) interaction between the API and the coformer.
In our case, the position of hydrogen is dynamic. In HIMO crystal structure, on the Fourier map there were peaks on the vicinity of both oxygen, so the hydrogen atoms were localized in this places, restrained and refined with partial occupancy (sum of fractional occupancy is 1). In both cocrystals BA/HIMO and TBA/HIMO the hydrogen is transferred from BA and TBA molecules and both oxygens connected to nitrogen. The equal N-O bond length, sp hybridization of carbon in BA and TBA at BA/HIMO and TBA/HIMO co-crystals support assignment to the group of cocrystal. The dynamic nature of these hydrogens is also visible on the NMR spectra.
The another criteria were published by Katie L. Cavanagh, Chinmay Maheshwari, Naír Rodríguez-Hornedo in paper “Understanding the Differences Between Cocrystal and Salt Aqueous Solubilities” Journal of Pharmaceutical Sciences 107 (2018) 113-120. Employing given supra criteria we did not find arguments for define samples obtained in this work as salts.
Line 334: I believe that is Figure 5 and not 4.
Answer. It is corrected in revised version.
Figure 7: The same question that I made for Figure 3 can also be made here.
Answer. Please see our answer to comment related with Figure 3. It is also valid for Figure 7.
Line 553: Why did the authors determined the solubility in ethanol and Ethyl Acetate? Those two solvents are not relevant for pharmaceutical application. Moreover, the solubility should be made in a biorelevant medium at a pH close to the one present in the site of absorption of the APIs. None of that is accounted for in this work.
Answer. The ethanol and ethyl acetate are often used in test of solubility of cocrystals. Below are selected papers confirming this statement.
- Samipillai, M.; Rohani, S. The Role of Higher Coformer Stoichiometry Ratio in Pharmaceutical Cocrystals for Improving their Solid-State Properties: The Cocrystals of Progesterone and 4-Hydroxybenzoic Acid. J. Cryst. Growth, 2019, 507, 270–282.
- Zong, S.; Pan, B.; Dang, L.; Wei, H. Stability, Solubility and Thermodynamic Properties of Dimorphs of Furosemide-4,4′-Bipyridine Cocrystals in Organic Solvents. J. Mol. Liq. 2019, 289 111017-11028.
- Lange, L.; Heisel, S.; Sadowski G. Predicting the Solubility of Pharmaceutical Cocrystals in Solvent/Anti-Solvent Mixtures. Molecules 2016, 21, 593-610.
- Alhalaweh, A.; Sokolowski, A.; Rodríguez-Hornedo, N.; Velaga, S. P. Solubility Behavior and Solution Chemistry of Indomethacin Cocrystals in Organic Solvents. Cryst. Growth Des. 2011, 11, 3923–3929.
- Good, D. J.; Rodrgíuez-Hornedo, N. Solubility Advantage of Pharmaceutical Cocrystals. Cryst. Growth Des. 2009, 9, 2253- 2264.
Moreover, according to the Food and Drug Administration, ethanol and ethyl acetate are even regarded as solvents of low toxic potential, so-called Class 3 solvents. (FDA-USA. Guidance for Industry—Q3C —Tables and List (Revision 4nd). In Administration, US Department of Health and Human Services, Food and Drug Administration ed.; Center for Drug Evaluation and Research Food and Drug Administration, FDA: Silver Spring, MD, USA, 2018.)
We agree with Reviewer that the solubility tests can be performed in different manners. The strategy chose in our project is one of the options.
Line 554: The authors state that solubility is important, however cocrystals can be used for other purposes. Do the APIs used in this work need protection? Also, if the objective was to improve solubility, the solubility values of the APIs should be stated in the introduction.
Answer. In most papers the increase of solubility is main motivation for preparation of pharmaceutical cocrystals. In our particular case the spectacular increase of solubility was not noted. Despite this, we believe that this does not disqualify HIMO as an active molecular partner in cocrystals formation. From a pharmaceutical point of view, HIMO is an interesting molecule with not fully discovered biological properties. As we highlighted in Introduction numerous imidazole derivatives are used as drugs. Maybe not in case of TBA and BA but in the case of other APIs tested in our Laboratory synergetic therapeutic effect can be expected. HIMO is coformer with broad, multifunction properties also protective.

Reviewer 2 Report
This work advances our understanding of HIMO as a useful coformer for the preparation of pharmaceutical cocrystals. The manuscript from Wróblewska and colleagues describes the detailed structural and conformational analysis of HIMO, TBA and BA cocrystals which would be crucial for the development of new drug delivery systems based on 1-hydroxy imidazole 3-oxide derivatives and would be a potential candidates for pharmaceutical application.
This is carefully performed study that provides a wealth of new and well-substantiated data . I have only a few minor suggestions.
- in the unit cell diagrams (Fig 11, 13 and 15), the length could be included.
- The figure resolution could be improved.
Author Response
Reviewer 2
This work advances our understanding of HIMO as a useful coformer for the preparation of pharmaceutical cocrystals. The manuscript from Wróblewska and colleagues describes the detailed structural and conformational analysis of HIMO, TBA and BA cocrystals which would be crucial for the development of new drug delivery systems based on 1-hydroxy imidazole 3-oxide derivatives and would be a potential candidates for pharmaceutical application.
This is carefully performed study that provides a wealth of new and well-substantiated data . I have only a few minor suggestions.
- in the unit cell diagrams (Fig 11, 13 and 15), the length could be included.
Answer; The requested parameters are given in Table 1. We would be appreciate if reviewer will accept such form of presentation.
- The figure resolution could be improved.
Answer; In revised version X-ray Figures are replaced, resolution is improved.
Reviewer 3 Report
Please find the attachment.

Author Response
Reviewer 3
The manuscript ‘Application of 1-Hydroxy-4,5-Dimethyl-Imidazole 3- Oxide as Coformer in Formation of Pharmaceutica Cocrystals’ reported the two cocrystals with barbituric and thiobarbituric acids. The findings of this paper are very interesting and the experimental support for their claims are really good. However, I have following concern before it gets accepted in this journal:
- In the section 2.8, what is the basis for selecting these three solvents for the solubility? If the solvent selection is based on the tautomers of TBA then what is the basis of using the same solvent system for BA? Would BA also give the similar type of different tautomers/polymorphs/pseudopolymorphs in different solvent you used? Please clarify this.
Answer. There is no straightforward correlation between polymorphic/tautomeric forms of TBA/BA and used solvents for crystallization. In paper cited in our manuscript (ref. 39) Chierotti, M.R.; Ferrero, L.; Garino, N.; Gobetto, R.; Pellegrino, L.; Braga, D.; Grepioni, F.; Maini, L. The Richest Collection of Tautomeric Polymorphs: The Case of 2-Thiobarbituric Acid. Chem. Eur. J. 2010, 16, 4347‒4358 different methods of preparation of TBA forms were shown. Unfortunately these methods are not useful for BA. Only limited number of tautomeric polymorphs of BA is known. In our work, papers reporting these studies are cited (ref. 40, 44, 45, 46)
The, water, ethanol and ethyl acetate are often used in test of solubility of cocrystals. Test of solubility in water is obvious. Below are selected papers justifying the use of ethanol and ethyl acetate as solvents
Samipillai, M.; Rohani, S. The Role of Higher Coformer Stoichiometry Ratio in Pharmaceutical Cocrystals for Improving their Solid-State Properties: The Cocrystals of Progesterone and 4-Hydroxybenzoic Acid. J. Cryst. Growth, 2019, 507, 270–282.
Zong, S.; Pan, B.; Dang, L.; Wei, H. Stability, Solubility and Thermodynamic Properties of Dimorphs of Furosemide-4,4′-Bipyridine Cocrystals in Organic Solvents. J. Mol. Liq. 2019, 289 111017-11028.
Lange, L.; Heisel, S.; Sadowski G. Predicting the Solubility of Pharmaceutical Cocrystals in Solvent/Anti-Solvent Mixtures. Molecules 2016, 21, 593-610.
Alhalaweh, A.; Sokolowski, A.; Rodríguez-Hornedo, N.; Velaga, S. P. Solubility Behavior and Solution Chemistry of Indomethacin Cocrystals in Organic Solvents. Cryst. Growth Des. 2011, 11, 3923–3929.
Good, D. J.; Rodrgíuez-Hornedo, N. Solubility Advantage of Pharmaceutical Cocrystals. Cryst. Growth Des. 2009, 9, 2253- 2264.
Moreover, according to the Food and Drug Administration, ethanol and ethyl acetate are even regarded as solvents of low toxic potential, so-called Class 3 solvents. (FDA-USA. Guidance for Industry—Q3C —Tables and List (Revision 4nd). In Administration, US Department of Health and Human Services, Food and Drug Administration ed.; Center for Drug Evaluation and Research Food and Drug Administration, FDA: Silver Spring, MD, USA, 2018.)
- Description for the figure 4 needs clarification or re-writing to clarify which tautomer is what instead of solvent system in the description or mention both.
Answer. It is done in revised version. We removed solvent labels and in legend of Figures we added comments defining tautomers.
- Also, from figure 4, do authors mean that; to prepare right side material d, e, f, they used a, b, c? I mean to prepare ‘d’, physical mixture ‘a’ is used? If not, please correct the description and explanation accordingly.
Answer. In the Legend of Figure 4
- is leading to d)
- is leading to e)
- is leading to f)
- In figure 7, I would put 13C CP/MAS spectrum of the coformer (BA), cocrystal (BA_HIMO) and HIMO to see the differences. Also, please put the stack image of the calculated PXRD pattern and the theoretical pattern of the cocrystal for better understanding of purity of bulk powder material.
Answer. Thank you for this comment. For better clarity we decided to add the Supporting Information. In SI we show the calculated diffractograms for three forms of TBA used in our project employing crystallographic data published by Chierotti et al. Diffractograms for TBA sample crystallized in acetonitrile, ethanol, water as well HIMO and TBA/HIMO are depicted in Figure S1. Figure S3 shows the PXRD calculated diffractograms for BA forms, HIMO and BA/HIMO. The crystallographic data are taken from papers; Lewis T. C, Tocher D. A., Price S. L. An Experimental and Theoretical Search for Polymorphs of Barbituric Acid: The Challenges of Even Limited Conformational Flexibility Crystal Growth & Design 2004, 4, 979-987 and Nichol G.S., Clegg W., Acta Crystallographica,Section B: Structural Science, A variable-temperature study of a phase transition in barbituric acid dihydrate 2005, 61, 464-472
- It seems Figures 12, 13, 14 and 15 are wrong as the HIMO has two hydrogens present of the oxygen atom bonded to both the nitrogens.
Answer. The position of mentioned hydrogens is dynamic. In HIMO crystal structure, on the Fourier map there were peaks on the vicinity of both oxygen, so the hydrogen atoms were localized in this places, restrained and refined with partial occupancy (sum of fractional occupancy is 1). In both co-crystals BA/HIMO and TBA/HIMO the hydrogen transferred from BA and TBA molecules and both oxygens connected to nitrogen are protonated. The equal N-O bond length ,sp hybridization of carbon in BA and TBA at BA/HIMO and TBA/HIMO co-crystals indicated that this interpretation is proper. The dynamic nature of hydrogens is also visible on the NMR spectra.
- Figure 17 represents which solvent system as you have done in three systems? Also, please clarify how many experiments have you done for this solubility study?
Answer. Concentrations determined for BA and TBA (in pure form and in the tested cocrystals) in three solvents: water, EtOH and AcOEt were reported as the average of two replicated experiments for each sample. The injections for each sample were repeated four times (eight results for each sample).
- I would like to know whether you have done any stability study on the powder obtained (after filtration) in this solubility study for 12 hours such as PXRD, DSC etc, in order to understand the cocrystal stability. It is important as the coformers are polymorphic/tautomeric.
Answer. In fact submitting manuscript we have not tested stability of powders after solubility experiments. The question of referee prompted us to carry out requested measurements. Because of difficult time related with coronavirus pandemia we have no access to all instruments and techniques employed in our project. Thus we have only performed the 13C CP/MAS measurements. The obtained data proves that powders are stable and do not undergo decomposition during the solubility tests. Only the traces of new crystallographic form were observed in the case of TBA/HIMO cocrystal dissolved in ethyl acetate.
- Line number 566 from the EHS point of view; authors have used MeOH as a solvent which is usually not desired solvent compared EtOH or water. Just this line is contradicting the solvent used in your study.
Answer. The methanol was used only on the stage of cocrystals preparation, as convenient solvent for crystallization (wet method) and as preferred LAG in grinding procedure. None of obtained cocrystals contained methanol in the crystal lattice. Thus the obtained cocrystals are methanol free.
Round 2
Reviewer 1 Report
The work as improved but some of the questions were not completely answered by the authors and I cannot recommend the acceptance of the paper until these question are answered.
In relation with the DSC thermograms present, the authors say that their compound behaves similar to some of the literature. I´m sorry but I disagree. In both papers cited by the authors, the DSC thermogram of the cocrystal present an endotherm peak correspondent to the melting and only after the exothermpeak from the degradation. Therefore I´m still intrigued by the behaviour of your compound, and no explanation for its behaviour was given by the authors. Moreover, it would be beneficial to show the thermograms of the pure compounds.
Regarding the solvents used for the dissolution, the authors claim that many others used in their work, a say that the solvents are not toxic. Both those reasons can be true, but the fact remains that both solvents are not biological relevant. If the in vitro activity is made with such solvents the correlation in vitro/in vivo cannot be evaluated because in-vivo those solvents are not present in our body. The dissolution should be made in aqueous simulated body fluids or at least in water at a pH close to the one present in the site of absorption of the API studied.
Author Response
The work as improved but some of the questions were not completely answered by the authors and I cannot recommend the acceptance of the paper until these question are answered.
1) In relation with the DSC thermograms present, the authors say that their compound behaves similar to some of the literature. I´m sorry but I disagree. In both papers cited by the authors, the DSC thermogram of the cocrystal present an endotherm peak correspondent to the melting and only after the exotherm peak from the degradation. Therefore I´m still intrigued by the behaviour of your compound, and no explanation for its behaviour was given by the authors. Moreover, it would be beneficial to show the thermograms of the pure compounds.
Answer. Maybe the literature examples presented in previous response were not selected properly and are not fully representative. The thermal properties of BA/HIMO and TBA/HIMO cocrystals are not unique. The very similar profile was also observed for TATB/HMX Cocrystal in paper (DOI: 10.1002/prep.201800004) Fabrication of Ultra-fine TATB/HMX Cocrystal Using a Compound Solvent by Conghua Hou, Yuanping Zhang, Yunge Chen, Xinlei Jia, Shi min Zhang, and Yingxin Tan, Propellants Explos. Pyrotech. 2018, 43, 1–8. Searching 1,3,5-triamino-2,4,6-trinitrobenzene (TATB), cyclotetramethylenetetranitramine (HMX) and cocrystal authors have not observed the endothermic processes (please see Figure 6 in cited paper, this Figure is also attached in material for reviewer). We agree that thermal properties of cocrystals under discussion require more advanced discussion and probably further studies. However, in our opinion such problem should be considered as a separate project. We hope that referee will accept our point of view. As requested we attach the DSC profiles of physical mixtures BA and HIMO as well TBA and HIMO. The HIMO melting point was measured by Hot-stage micrcoscopic technique and was establish to be 200 °C. The melting point for BA and TBA according to literature is 245 °C.
2) Regarding the solvents used for the dissolution, the authors claim that many others used in their work, a say that the solvents are not toxic. Both those reasons can be true, but the fact remains that both solvents are not biological relevant. If the in vitro activity is made with such solvents the correlation in vitro/in vivo cannot be evaluated because in-vivo those solvents are not present in our body. The dissolution should be made in aqueous simulated body fluids or at least in water at a pH close to the one present in the site of absorption of the API studied.
Answer; We fully agree with interpretation of referee. Maybe our intention for showing broader spectrum of physicochemical properties of obtained cocrystals, including solubility in different solvents is not consistent with major subject of this work. Thus in revised version we decided significantly change the part of manuscript describing solubility studies. First we removed data for ethyl acetate. Second we carried out new measurements. It was not easy in corona virus time because our Institute is already three weeks closed and special permission was necessary for short stay in Lab. The new results show solubility of BA, TBA, BA/HIMO and TBA/HIMO in simulated gastric fluid without pepsin, SGFsp at pH 1.2. SGFsp was prepared by taking 2g NaCl and 7 mL concentrated HCl in 1000 mL distilled water. This information is added in Experimental Section (page 5). Figure 17 is replaced be the new one, description of results in page 20 is consistent with Figure 17 (page 21)

Round 3
Reviewer 1 Report
I would like to thank the authors for the effort on doing further experiments during these difficult days.
The paper is improved and can be published as it is.